# *Artemisia annua* Extract Attenuate Doxorubicin-Induced Hepatic Injury via PI-3K/Akt/Nrf-2-Mediated Signaling Pathway in Rats

**DOI:** 10.3390/ijms242115525

**Published:** 2023-10-24

**Authors:** Karim Samy El-Said, Ahmed S. Haidyrah, Maysa A. Mobasher, Arwa Ishaq A. Khayyat, Afnan Shakoori, Noorah Saleh Al-Sowayan, Ibrahim Omar Barnawi, Reham A. Mariah

**Affiliations:** 1Biochemistry Division, Chemistry Department, Faculty of Science, Tanta University, Tanta 31527, Egypt; kareem.ali@science.tanta.edu.eg; 2Digital & Smart Laboratories (DSL), King Abdulaziz City for Science & Technology (KACST), Riyadh 11442, Saudi Arabia; ahydrah@kacst.edu.sa; 3Department of Pathology, Biochemistry Division, College of Medicine, Jouf University, Sakaka 72388, Saudi Arabia; mmobasher@ju.edu.sa; 4Biochemistry Department, Science College, King Saud University, Riyadh 11451, Saudi Arabia; aalkhyyat@ksu.edu.sa; 5Laboratory Medicine Department, Faculty of Applied Medical Sciences, Umm Al-Qura University, Makkah 21955, Saudi Arabia; amshakoori@uqu.edu.sa; 6Department of Biology, College of Science, Qassim University, Buraydah 52377, Saudi Arabia; 7Department of Biological Sciences, Faculty of Science, Taibah University, Al-Madinah Al-Munawwarah 41321, Saudi Arabia; abrnawy@taibahu.edu.sa; 8Department of Medical Biochemistry, Faculty of Medicine, Tanta University, Tanta 31527, Egypt

**Keywords:** *Artemisia annua*, phytochemicals, antioxidants, doxorubicin, hepatotoxicity

## Abstract

Doxorubicin (DOX), which is used to treat cancer, has harmful effects that limit its therapeutic application. Finding preventative agents to thwart DOX-caused injuries is thus imperative. *Artemisia annua* has numerous biomedical uses. This study aims to investigate the attenuative effect of *Artemisia annua* leaf extract (AALE) treatment on DOX-induced hepatic toxicity in male rats. A phytochemical screening of AALE was evaluated. Forty male rats were used; G1 was a negative control group, G2 was injected with AALE (150 mg/kg) intraperitoneally (i.p) daily for a month, 4 mg/kg of DOX was given i.p to G3 once a week for a month, and G4 was injected with DOX as G3 and with AALE as G2. Body weight changes and biochemical, molecular, and histopathological investigations were assessed. The results showed that AALE contains promising phytochemical constituents that contribute to several potential biomedical applications. AALE mitigated the hepatotoxicity induced by DOX in rats as evidenced by restoring the alterations in the biochemical parameters, antioxidant gene expression, and hepatic histopathological alterations in rats. Importantly, the impact of AALE against the hepatic deterioration resulting from DOX treatment is through activation of the PI-3K/Akt/Nrf-2 signaling, which in turn induces the antioxidant agents.

## 1. Introduction

Chemotherapeutic medications are frequently used to treat cancer. Despite their effectiveness, there are related negative consequences on the vital organs, so it is necessary to find novel ways to lessen these side effects [1]. Doxorubicin (DOX) is an antineoplastic drug that is frequently used to treat hematological malignancies, breast cancer, and sarcomas [2]. Although DOX has an anticancer effect, its use is restricted due to its negative impacts on non-cancerous cells [3]. DOX works by damaging the DNA of cancer cells, which stops them from multiplying, while it causes oxidative stress in non-cancerous cells and reduces mitochondrial activity [4]. The metabolism of DOX occurs through liver microsomal enzymes, which in turn leads to the buildup of harmful intermediates that have been linked to the development of liver damage [5]. DOX could promote oxidative stress via suppressing Nrf-2, which mediates cellular redox homeostasis and antioxidant responses by inducing the antioxidant response elements (ARE) of antioxidant genes [6]. A protein kinase Akt is a crucial regulator for apoptosis and survival and is activated via phosphoinositide 3-kinase (PI3-K)-dependent events. Reactive oxygen species (ROS), which promote p38, deactivate pAKT signaling, and cause apoptosis, have been found to be closely related to chemotherapy [7]. In addition to reducing oxidative stress and cell injury, activating the PI3-K/Akt signaling pathway can also suppress autophagy and apoptosis [8]. Alternative medicine is thought to be one strategy that could reduce the harmful effects of chemotherapy. Natural constituents provide promising antioxidants that could reduce DOX-induced liver malfunctions. For instance, naringenin ameliorated DOX-induced hepatotoxicity by reducing the oxidative stress [9]. Furthermore, the treatment with diosmin showed good impacts against DOX-induced hepatotoxicity via restoration of hepatic antioxidant factors [10].

*Artemisia annua* L. (*A. annua*), a member of the *Asteraceae* family, is a well-known medicinal herb used as an anti-malaria, anthelmintics, anti-diabetes, antioxidation, anti-inflammation, and immunomodulation agent [11,12,13]. The bioactive ingredients contained in *A. annua* and their antitumor activities have been reported via increased cell percentages in the S and G2/M phases, and caspase 3 activation [14]. In addition, *A. annua* are rich in flavonoids, polysaccharides, terpenoids, sterols, coumarins, acetylenes, and other bioactive compositions [15,16,17]. Furthermore, a previous study reported that the biologically active constituents in *A. annua* could provide potential protective effects against DOX-induced cardiotoxicity in experimental animals [18]. The administration of *A. judaica* extract alone or in combination with cyclophosphamide presented a significant attenuation of hepatic carcinoma in male rats [19]. Due to the limited evidence about the effects of *A. annua* on DOX-induced hepatotoxicity, the current study investigated the possible mechanism of the treatment with *Artemisia annua* leaf extract (AALE) against liver injury that was induced by DOX treatment in rats.

## 2. Results

### 2.1. Phytochemical Content of Artemisia annua Leaves

Quantitative phytochemical analysis of *Artemisia annua* leaves (AAL) showed that the total phenolic and flavonoid contents in the AAL were 21.36 ± 0.85 mg GAE/g DW and 69.86 ± 2.37 mg QUE/g DW, respectively. The total antioxidant capacity (TAC) of the AAL was 287.65 ± 5.54 mg AE/g DW. Saponin and anthocyanin contents were 415.28 ± 3.95 mg/g DW 3.98 ± 0.45 mg ECG/g DW, respectively. The DPPH scavenging activity (%) of AAL was 81% ± 1.57, and the concentration of AAL able to inhibit 50% of DPPH was 4.55 ± 0.34 mg/mL (Table 1).

### 2.2. GC-MS Analysis of Artemisia annua Leaf Extract

The gas chromatography-mass spectrometry (GC-MS) technique was used to analyze the bioactive compounds present in AALE. The results showed that 2,4,6-trimethyl-1,3,6-heptatriene was the most abundant compound, followed by 3-butenoic acid, 2-oxo-4-phenyl, β-caryophyllenea, α-santonin, spathulenol, and 3,5-heptadien-2-ol, 2,6-dimethyl. These phytochemicals’ peak area percentages (P.A %) were 11.37, 9.81, 8.95, 7.79, 6.83, and 6.57, respectively. The retention times (RTs) were 13.47, 16.21, 15.63, 25.39, 22.37, and 13.19, respectively (Figure 1 and Table 2).

### 2.3. Effect of AALE Treatment on the Percentages of Body Weight Changes

The results showed that there was significant decrease (*p* ≤ 0.05) in the percentage of body weight change in the DOX-injected group, up to 2.35%, when compared to the control groups. However, treatment of DOX/AALE led to significant improvement in the % BW, up to 18.29%, compared with the DOX-injected group (Figure 2).

### 2.4. Effect of AALE Treatment on the Alterations of Serum Biochemical Parameters Induced by DOX

Liver transaminases (AST and ALT) were significantly elevated (*p* ≤ 0.05) in the sera of the DOX-intoxicated group, reaching 97.58 ± 2.46 and 67.78 ± 1.44 U/L, respectively, when compared to the normal and AALE control groups. DOX/AALE-treated rats showed a considerable drop in the levels of AST and ALT, up to 56.39 ± 1.98 and 34.89 ± 1.16 U/L, respectively (Table 3). Rats that were injected with DOX showed a significant increase (*p* < 0.05) in alkaline phosphatase (ALP) serum level, at 511.8 ± 5.86 U/L, when compared to their control groups (287.6 ± 4.04 and 281.2 ± 2.95 U/L); however, the treatment with AALE caused a significant decrease in serum ALP level as compared to the DOX-intoxicated group (347.2 ± 6.49 U/L). Furthermore, the results showed that there were significant increases (*p* < 0.05) in the total and direct bilirubin of the DOX-injected group. Treating DOX-intoxicated rats with AALE improved the total and direct bilirubin serum levels (Table 3).

### 2.5. Treatment with AALE Modulated the Alterations of Hepatic Biochemical Parameters Induced by DOX

Hepatic MDA level was significantly increased (*p* < 0.05) in the DOX-injected group, by 2.5-fold compared to the control groups (Figure 3A). The GSH level was significantly decreased in the DOX-intoxicated group, by −2.7-fold compared to the control groups (Figure 3B). The group of rats that were treated with DOX/AALE showed a significant decrease in the MDA level compared to the DOX-treated group (0.305 ± 0.023 versus 0.579 ± 0.018 nmol/mg protein). However, treatment with DOX/AALE led to a significant increase in the GSH level as compared to the DOX-treated group (17.19 ± 0.94 versus 9.16 ± 0.54 mg/mg protein) (Figure 3).

As shown in Figure 4A, the intraperitoneal injection of DOX in rats was accompanied by a remarkable reduction in hepatic PI-3K to 281 pg/mg protein versus control (760.52 ± 29.56 pg/mg protein) and AALE control groups (775.33 ± 25.78 pg/mg protein). Furthermore, the results reported a 2.1-fold decrease in the hepatic Akt of DOX-treated rats when compared to the control group (21.61 ± 1.89 versus 45.14 ± 2.96 pg/mg protein). This reduction has been improved by the treatment with AALE (37.79 ± 3.14 pg/mg protein) (Figure 4B). Similarly, hepatotoxicity induced by DOX injection demonstrated a 1.9-fold reduction in hepatic Nrf-2 concentration versus control (273.92 ± 3.91 versus 140.24 ± 3.27 pg/mg protein). In the DOX-injected group (139 ± 3.58 pg/mg protein), rats treated with AALE showed a significant increase (*p* < 0.05) in hepatic levels of Nrf-2, as their levels reached 211.82 ± 5.79 pg/mg protein (Figure 4C). Hepatic HO-1 levels were markedly increased in the DOX/AALE-administrated group, almost close to the normal levels (Figure 4D).

### 2.6. Effect of AALE Treatment on Antioxidant Gene Expression

The relative gene expression levels of SOD, CAT, GPX, GST, and GR were determined to investigate the impact of DOX and/or AALE treatments on the gene expression of the antioxidants in liver tissues of rats. The molecular analysis showed that there were significant downregulations (*p* < 0.05) in the hepatic SOD and CAT genes by around 0.5-fold compared to the control groups; however, the treatment of DOX-intoxicated rats with AALE led to significant upregulation of these genes, almost close to the control groups (Table 4). As compared to the control groups, DOX-injected rats showed significant downregulation in the GPX, GST, and GR genes. Interestingly, rats treated with DOX/AALE showed marked up-regulation in the hepatic GPX, GST, and GR genes (Table 4).

### 2.7. Treatment with AALE Restored Hepatic Histopathological Alterations Induced by DOX

The histopathological investigations of H&E-stained liver sections from the negative control group showed normal hepatocyte lobulation with alternated blood sinusoids lined by an endothelial cell layer. Hepatocytes revealed centrally located nuclei (Figure 5A). The liver section of rats administered with AALE showed normal hepatic architecture, almost regular central veins with activated Kupffer cells were noticed (Figure 5B). However, the liver section of rats injected with DOX displayed severe congested and dilated central vein; hepatocytes were mostly degenerated with vacuolated cytoplasm, and nuclear changes such as pyknotic nuclei and disrupted blood sinusoids were observed, along with cellular infiltrations (Figure 5C). The liver section of the group treated with DOX/AALE exhibited an improvement in the hepatic architectures, represented by regular central vein, with fewer congestions, few numbers of binucleated hepatocytes, less cellular infiltrations, and almost normal Kupffer cells (Figure 5D).

## 3. Discussion

Although chemotherapy is helpful in treating cancer, side effects limit its usefulness in tumor therapy [20]. One of the most effective anthracycline-derivative anticancer medicines is DOX; however, its therapeutic use has been severely constrained due to toxicity [21]. Liver is a vulnerable organ to DOX damage. Therefore, it is important to find effective and safe agents against the hepatotoxicity of DOX. Antioxidant agents alleviate hepatotoxicity via adjustment of oxidative stress and apoptosis [22]. The *A. annua* is well-known for its medicinal properties in preventing many ailments. Tue bioactive constituents of *A. annua* are valuable sources for developing pharmaceuticals [23]. The defensive capacity of *A. annua* as a powerful antioxidant cure against free radical harm has been detailed [24]. The present study addressed the novel impact of AALE versus the hepatotoxicity induced by DOX in rats via the PI-3K/Akt/Nrf-2-mediated signaling pathway.

The phytochemistry and the valuable pharmacological activity of *Artemisia* were reviewed by a previous study by Bisht et al. (2021) [25]. This study showed that the AALE contains adequate promising contents of phenolics, flavonoids, saponins, and anthocyanins as well as DPPH scavenging activity. The antioxidant effect of AALE is related to its phytochemical compounds, which contribute to its biomedical potential [26]. These findings agree with previous studies that screened the phytochemical profiles of *Artemisia* [27,28,29]. In the current study, GC-MS analysis for AALE showed that about seventeen potent compounds were detected; these compounds belonged to terpene, aliphatic, and aromatic compounds. It has been reported that the AALE was found to contain steroids and terpenoids, with several phyto-constituents screened by the GC-MS method [30]. For instance, 2,4,6-trimethyl-1,3,6-heptatriene was detected as a bioactive agent in AALE, which agreed with Liu et al. (2022), who detected the same constituent as the main component in AALE [31]. Interestingly, in this study, three natural sesquiterpenes, β-caryophyllenea, α-Santonin, and spathulenol, were abundantly identified in AALE, which provide several biological activities, including antioxidant, antimicrobial, anticarcinogenic, and anti-inflammatory effects [32,33,34].

This study revealed that the DOX-injected group showed a significant decrease in their % BW; this could be due to the toxic effects of DOX on vital tissues and metabolic disorders, in agreement with previous studies that established the impact of DOX administration on the BW of experimental animals [35,36]. They attributed these adverse effects of DOX on the BW to decreased appetite, reduced feed intake, disruption of basal metabolism, and inhibition of protein production caused by DOX toxicity. Treatment with DOX/AALE led to significant improvement in the % BW; this finding agreed with previous reports demonstrating the effect of medicinal products on the improvement of body weight loss promoted by DOX injection [37,38]. Sera AST, ALT, and ALP activities as well as bilirubin levels were significantly elevated in the DOX-intoxicated rats due to hepatocellular damage, loss of hepatic functioning, and leakage to the bloodstream from necrotic hepatocytes, because of DOX-induced toxicity; however, treatment of DOX-injected rats with AALE caused a significant reduction in these parameters. This could be due to the improvement of hepatic functioning induced by AALE treatment. High levels of blood markers for both hepatocellular and hepatobiliary degenerations and the ameliorative effects of natural products were previously reported [39,40].

The DOX has been shown to cause an imbalance of redox status and attenuation of the antioxidants that led to the oxidization of macromolecules, leading to liver injuries [10,39,40,41]. Allied with previous results, the study showed a significant increase in the MDA levels with a reduction in GSH levels herein in DOX-injected rats; this could be assigned to the impaired hepatic tissue under the damaging effects of DOX. Jagetia and Lalrinengi (2017) reported that due to its ability to increase MDA and deplete GSH, DOX may increase liver damage [42]. Concomitant administration of AALE with DOX led to a reduction in the hepatic MDA and an increase in GSH concentration to relieve the oxidative stress and restore antioxidant/oxidant hemostasis, which is a key factor in suppressing liver injury induced by DOX. This is consistent with previous publications that reported the impact of natural constituents on oxidative stress induced by DOX [43,44,45]. The decreased hepatic HO-1 and Nrf-2 protein has been implicated in the process of DOX-induced liver injury [46]. In our work, rats administered with DOX showed significant decreases in the hepatic Nrf-2 and HO-1 levels; these reductions were reversed by AALE treatment, suggesting the impact of these herbs in mitigating oxidative stress caused by DOX. The modulation of the PI-3K/Akt signaling pathway regulates cell development; numerous downstream proteins are activated by phosphorylated Akt [47]. In the present study, DOX administration led to the suppression of Nrf-2, PI-3K, and Akt levels in liver tissues of rats. Treatment with DOX/AALE for 30 days restored hepatic Nrf-2, PI-3K, and Akt levels. These findings highlight the attenuative effect of AALE on the hepatotoxicity brought on by DOX via Nrf2/HO-1 activation and PI-3K/Akt axis modulation.

This study reported that administration of DOX to rats led to significant downregulations in the hepatic antioxidant genes. These results were in accordance with a previous report indicating that DOX triggers oxidative stress by building up ROS in animals’ liver and decreases antioxidant gene expression [48]. Interestingly, the supplementation of DOX-administered rats with AALE resulted in the upregulation of the antioxidant gene expression, which agreed with previous reports demonstrating the impact of natural products on gene expression of antioxidant genes upon the administration of DOX [9,49]. A previous report stated that DOX-treated rats can increase liver enzyme activity and histological alterations due to liver injury [49]. In the current study, the histopathological investigations of liver sections from DOX-injected rats showed severe congested central vein and degenerated hepatocytes; this finding was in accordance with Mete et al.’s (2016) results, which reported necrosis of biliary ducts in the DOX-treated group [50]. However, these histological alterations in the hepatic structures were dramatically restored in the rats that were concomitantly treated with DOX/AALE, which exhibited an improvement in the hepatic architectures, represented by regular central vein, fewer congestions, and less cellular infiltrations, implying protection from DOX-induced hepatic damage. These findings are in line with previous studies that reported the ameliorative effects of natural products against DOX-induced hepatotoxicity in rats [50,51,52].

## 4. Materials and Methods

### 4.1. Chemicals

Doxorubicin was purchased from Hikma Pharmacy Company (Cairo, Egypt). Aluminum chloride, potassium acetate, lead acetate, phenol, and sodium nitroprusside were purchased from Merck Company (Darmstadt, Germany).

### 4.2. Collection of Plant and Extract Preparation

The leaves of *A. annua* were obtained from the Crop Institute Agricultural Research Center, Giza, Egypt. The plant was authenticated and complied with relevant institutional guidelines. A specialist identified *A. annua* leaves, and a specimen was placed in the Tanta University Herbarium (code #TAN-147). The leaves were pulverized after being shade dried. A mixture of 500 mL 70% ethanol and 50 g of leaves powder was used; after filtering, the *Artemisia annua* leaf extract (AALE) was dried.

### 4.3. Determination of Phytochemical Content of A. annua Leaves

Contents of the total phenolic, flavonoid, and the total antioxidant capacities (TACs) in AALE were determined [53,54,55]. Saponin and anthocyanin were estimated according to Hiai et al. (1975) [56]. The DPPH radical scavenging capacity was spectrophotometrically evaluated [57].

### 4.4. Gas Chromatography and Mass Spectrum (GC-MS) Profiling of AALE

The phytoconstituents of AALE were determined using Trace GC 1310-ISQ mass spectrometer “GC-MS” (Thermo Scientific, Austin, TX, USA). By comparing the components’ retention times and mass spectra to those in the WILEY 09 and NIST 11 mass spectral databases, the components were identified.

### 4.5. Rats and Experimental Design

Forty male Sprague Dawley rats (130–150 g) were purchased from Alexandria University, Egypt. The animals were housed and maintained for a week’s acclimatization period. Our research was conducted in accordance with the standards endorsed by the Tanta University Faculty of Science Research Ethical Committee (IACUC-SCI-TU-0321). Rats were equally divided into four groups. G1 was injected i.p with saline in a proportional volume and in the same time scheme of other experimental groups; G2 was injected with AALE (150 mg/kg) i.p. daily for a month [58]; G3 was injected with 4 mg/kg of DOX i.p once a week for a month [59]; G4 was injected with DOX as G3 and administered with AALE as G2. The percentage of body weight changes (% b.wt) was calculated. All groups had been anesthetized by isoflurane, blood samples were collected by cardiac puncture, and then sera were separated for biochemical assays. Liver tissues were collected in ice-cold saline and homogenized for other biochemical and gene expression analysis. Furthermore, liver tissues were sectioned in buffered formalin for investigations into histopathology.

### 4.6. Biochemical Analysis

Serum levels of aspartate aminotransferase (AST) (catalog no. AS106145), alanine aminotransferase (ALT) (catalog no. AL103145), alkaline phosphatase (ALP) (catalog no. AP1020), total bilirubin (catalog no. BR1111), and direct bilirubin (catalog no. BR1112) were assessed using colorimetric kits (Spectrum Diagnostics, Egypt). Hepatic malondialdehyde (MDA) and reduced glutathione (GSH) levels (catalog no. MD2529 and catalog no. GR2511, respectively) were measured by using their kit (Biodiagnostic, Egypt). Liver homogenates were used for determination of phospho-PI-3 kinase by using their ELISA kit (catalog no. ab207485) (Abcam, Boston, MA, USA). Hepatic rat’s phospho-Akt concentrations were assessed by rat’s pAkt (Ser473) ELISA kit (catalog no. MBS775153). Furthermore, nuclear Nrf-2 levels in liver were evaluated using rat’s ELISA kits from MyBioSource, Inc., San Diego CA, USA (catalog no.: MBS752046). Hemoxygenase-1 (HO-1) levels in the liver were calculated using a rat-specific ELISA kit from Elabscience Biotechnology Inc., Houston TX, USA (catalogue no.: E-EL-R0864).

### 4.7. Molecular Analysis

Gene expression analysis was performed by using SYBR Green to assess mRNA expression of antioxidant genes in the tissues of the liver, using β-actin as a reference. The primers used are shown in Table 5, their sequence similarity was checked with BLAST (www.ncbi.nlm.nih.gov/blast/Blast.cgi accessed on April 2023).

### 4.8. Histopathological Investigations

Formalin-fixed liver sections were embedded in paraffin blocks after being processed in different alcohol and xylene grades. To investigate severe cellular damage, sections (5 μm) were produced, stained with hematoxylin and eosin, and examined under light microscope (Optika light microscope (B-350) [60].

### 4.9. Statistical Analysis

Graph Pad Prism software (San Diego, CA, USA) was used to analyze the one-way ANOVA results, followed by Tukey’s test for multiple comparisons, acceptable significance was recorded when *p* ≤ 0.05.

## 5. Conclusions

The AALE could be a potent beneficial agent against hepatotoxicity resulting from DOX treatment. This effect of AALE is through the activation of PI-3K/Akt/Nrf-2 signaling pathway, which in turn induces antioxidant enzymes. The combination of AALE and DOX may serve as a new strategy for efficient chemotherapy. Further studies should investigate the effects of AALE against other toxicities induced by DOX on the vital organs and their different mechanisms.

## Figures and Tables

**Figure 1 ijms-24-15525-f001:**
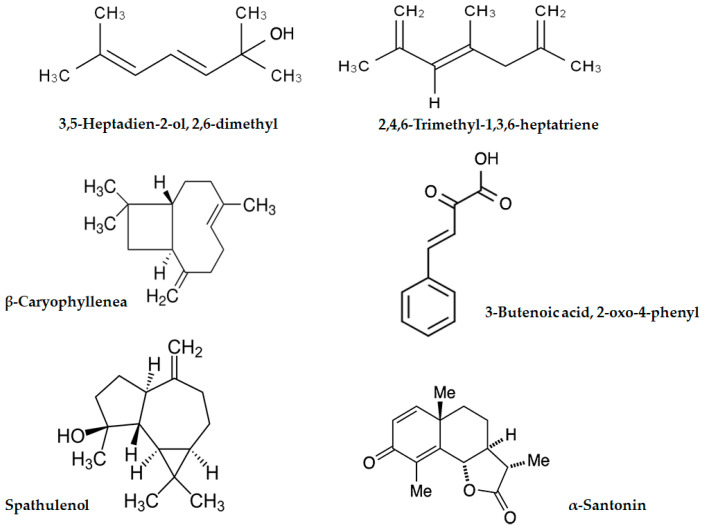
The phytochemical components that are most prevalent in AALE.

**Figure 2 ijms-24-15525-f002:**
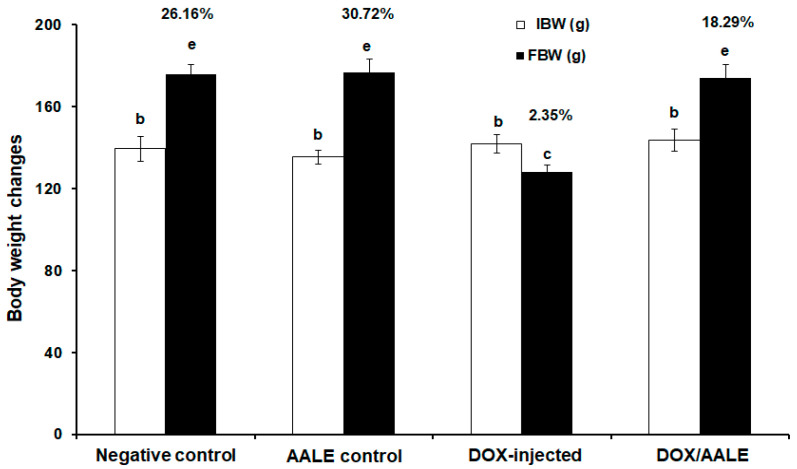
The initial and final body weights and their percentages of change in the different groups. Means that do not share a letter were significantly different (*p* < 0.05).

**Figure 3 ijms-24-15525-f003:**
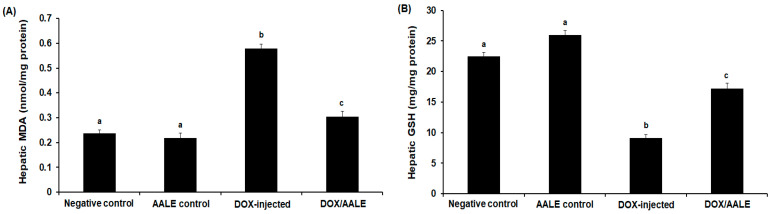
(**A**) Hepatic MDA and (**B**) GSH levels in the different groups. The values represented means ± S.D. Means that do not share a letter were significantly different (*p* < 0.05).

**Figure 4 ijms-24-15525-f004:**
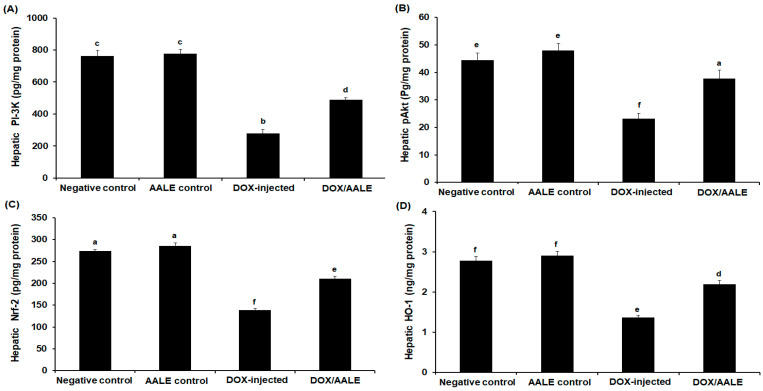
(**A**) Hepatic phosphatidylinositol 3-kinase (PI-3K), (**B**) hepatic protein kinase B (Akt), (**C**) hepatic nuclear factor erythroid 2-related factor 2 (Nrf-2), and (**D**) hepatic hemoxygenase (HO-1) levels in the different groups. The values represent means ± S.D. Means that do not share a letter were significantly different (*p* < 0.05).

**Figure 5 ijms-24-15525-f005:**
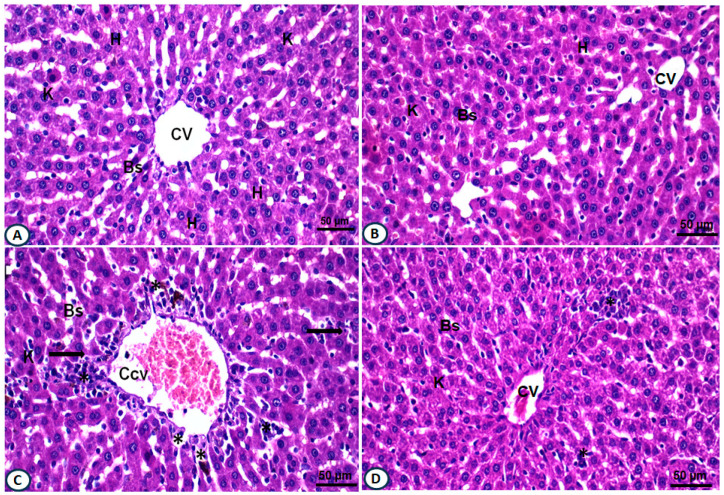
(**A**) Photomicrograph of liver section of normal control group shows normal hepatic structure: regular central veins (CV), normal hepatocytes (H), with normal blood sinusoids (Bs) and Kupffer cells (K). (**B**) Liver section of AALE control group shows mostly noral hepatocytes, with normal Bs and K (**C**). Liver section of DOX-injected group exhibits disorganization of hepatic architecture, severe congestion and dilated CV, cellular infiltrations (*), vacuolated cytoplasm (V), pyknotic nuclei (arrows), with irregular blood sinusoids and distinct K. (**D**) Liver section of DOX/AALE-treated group shows improvement in the hepatic organization, represented by fewer congestions in the central vein, few binucleated hepatocytes, less cellular infiltrations, and almost normal Kupffer cells (X 400).

**Table 1 ijms-24-15525-t001:** Quantitative phytochemical analysis of AAL.

Phytochemical Analysis	AAL
Total phenolic (mg GAE/g DW)	21.36 ± 0.85
Total flavonoids (mg QE/g DW)	69.86 ± 2.37
TAC (mg AAE/g DW)	287.65 ± 5.54
Saponin (mg/g DW)	415.28 ± 3.95
Anthocyanin (mg ECG/g DW)	3.98 ± 0.45
DPPH scavenging %	81% ± 1.57
IC_50_ of DPPH (mg/mL)	4.55 ± 0.34

DW: Dry weight, GAE: Gallic acid equivalent, QE: Quercetin equivalent. TAC: Total antioxidant capacity, ECG: Epicatechin gallate, AAE: Ascorbic acid equivalent.

**Table 2 ijms-24-15525-t002:** GC-MS analyses of *Artemisia annua* leaf extract (AALE).

No.	RT (min)	Name	M.F.	P.A%
1	3.65	1,2-15,16-Diepoxyhexadecane	C_16_H_30_O_2_	3.65
2	4.25	3,5-Hexadien-2-ol2-methyl	C_7_H_12_O	3.93
3	5.14	Cholestan -3-ol,2 methylene, (3ß,5α)	C_28_H_48_O	1.78
4	8.75	3,5-Heptadienal,2- ethylidene-6-methyl	C_10_H_14_O	3.87
5	9.27	Exo-2,7,7-trimethylbicyclo[2.2.1] eptan-2-ol	C_10_H_18_O	2.30
6	12.02	2-Cyclohexen-1-one, 3-methyl-6-(1-methylethyl)	C_10_H_16_O	2.65
7	13.19	3,5-Heptadien-2-ol, 2,6-dimethyl	C_9_H_16_O	6.57
8	13.47	2,4,6-Trimethyl-1,3,6-heptatriene	C_10_H_16_	11.37
9	15.63	β-Caryophyllenea	C_10_H_24_	8.95
10	16.21	3-Butenoic acid, 2-oxo-4-phenyl	C_10_H_8_O_3_	9.81
11	17.59	1,3,3-Trimethyl-2-oxabicyclo[2.2.2]octane	C_10_H_18_O	2.78
12	22.37	Spathulenol	C_15_H_24_O	6.83
13	25.00	Phytol	C_20_H_40_O	3.54
14	25.39	α-Santonin	C_15_H_18_O_3_	7.79
15	26.04	Naphtho[1,2-b]furan-2,6(3H,4H)-dione,3a,5,5a,9, 9a,9b-Hexahydro-9-hydroxy-3,5a,9-trimethyl	C_15_H_20_O_4_	3.25
16	27.33	1-Naphthalenecarboxylic acid, 5,6,7,8-tetrahydro	C_11_H_12_O_2_	1.32
17	29.72	Propanedioic acid, (phenylmethyl)-, diethyl ester	C_14_H_18_O_4_	2.74

RT: Retention time, M.F.: Molecular formula, P.A%: Peak area percentage.

**Table 3 ijms-24-15525-t003:** Serum aspartate transaminase (AST), alanine transaminase (ALT), alkaline phosphatase (ALP), total bilirubin (T.B), and direct bilirubin (D.B) in different groups.

Groups	AST (U/L)	ALT (U/L)	ALP (U/L)	T.B. (mg/dL)	D.B. (mg/dL)
Normal control	34.17 ± 1.15 ^f^	23.43 ± 0.59 ^a^	287.6 ± 4.04 ^e^	0.68 ± 0.014 ^c^	0.136 ± 0.005 ^a^
AALE control	30.64 ± 0.83 ^f^	20.21 ± 0.73 ^a^	281.2 ± 2.95 ^e^	0.60 ± 0.011 ^c^	0.129 ± 0.008 ^a^
DOX-treated	97.58 ± 1.46 ^b^	67.78 ± 1.44 ^e^	511.8 ± 5.86 ^b^	1.57 ± 0.056 ^b^	0.417 ± 0.009 ^d^
DOX/AALE	56.39 ± 1.68 ^c^	34.89 ± 0.96 ^a,d^	347.2 ± 6.49 ^f^	0.92 ± 0.015 ^c,f^	0.251 ± 0.007 ^e^

The values represent mean ± S.D. Number of rats was 10 per group. Means that do not share a letter in each column were significantly different (*p* < 0.05).

**Table 4 ijms-24-15525-t004:** Molecular analysis showed the fold change of superoxide dismutase (SOD), catalase (CAT), glutathione peroxidase (GPX), glutathione-S-transferase (GST), and glutathione reductase (GR) genes in different groups.

Groups	SOD	CAT	GPX	GST	GR
Normal control	1.01 ± 0.05 ^e^	1.00 ± 0.06 ^a^	1.02 ± 0.05 ^d^	1.00 ± 0.07 ^f^	1.00 ± 0.06 ^c^
AALE control	1.12 ± 0.03 ^e^	1.25 ± 0.04 ^a^	1.30 ± 0.06 ^d^	1.28 ± 0.05 ^f^	1.31 ± 0.04 ^c^
DOX-treated	0.46 ± 0.08 ^b^	0.53 ± 0.05 ^f^	0.37 ± 0.04 ^e^	0.41 ± 0.06 ^b^	0.50 ± 0.009 ^a^
DOX/AALE	0.85 ± 0.06 ^e^	0.91 ± 0.06 ^a^	0.78 ± 0.09 ^d^	0.92 ± 0.05 ^f^	0.74 ± 0.007 ^a,c^

The values represent mean ± S.D. Number of rats was 10 per group. Means that do not share a letter in each column were significantly different (*p* < 0.05).

**Table 5 ijms-24-15525-t005:** Forward and reverse primer sequences.

Gene	Accession Number	Forward Sequence (5′-3′)	Reverse Sequence (5′-3′)
SOD	NM_017050	CGAGCATGGGTTCCATGTC	CTGGACCGCCATGTTTCTTAG
CAT	NM_012520.2	ACAACTCCCAGAAGCCTAAGAATG	GCTTTTCCCTTGGCAGCTATG
GPX	NM_030826.4	GGAGAATGGCAAGAATGAAGA	CCGCAGGAAGGTAAAGAG
GST	XM_343545.8	GCTGGAGTGGAGTTTGAAGAA	GTCCTGACCACGTCAACATAG
GR	NM_053906.2	TTCTGGAACTCGTCCACTAGG	CCATGTGGTTACTGCACTACTTCC
β-actin	NM_031144.3	ATCGCTGACAGGATGCAGAAG	AGAGCCACCAATCCACACAGA

## Data Availability

Data sharing not applicable.

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
