# Peer review of "Artemisia annua Extract Attenuate Doxorubicin-Induced Hepatic Injury via PI-3K/Akt/Nrf-2-Mediated Signaling Pathway in Rats"

_ijms, 2023, doi:10.3390/ijms242115525_

Round 1

Reviewer 1 Report

The article proposed by Karim Samy El-Said et al., evaluate the effect of Artemisia Annua extract to attenuate doxorubicin-induced hepatic injury.

The manuscript is well written and clear.

I have some questions before publication:

Did the authors assess cardiotoxicity in these animals? If so, the results should be presented.

AALE is a mixture of different components (as shown in table 2). Do the authors have any idea which of these components might have a beneficial effect?

What do authors want write with: “means that do not share a letter were significantly different” ? Between which groups were the statistical comparisons made? The number of animals/group should appear in the legend.

Author Response

We thank the reviewer for his questions and suggestions that increase the quality of this paper. We hope the changes that have been made were appropriate and the manuscript can now be accepted for publication.

Reviewer 2 Report

The manuscript entitled “Artemisia Annua Extract Attenuate Doxorubicin-Induced Hepatic Injury via PI-3K/Akt/Nrf-2-Mediated Signaling Pathway in Rats” authored by El-Said and co-workers describes Artemisia Annua leaves extract acts as beneficial agent on the hepatotoxicity resulted from DOX treatment in rats. This manuscript explains the strategy for the combination of Artemisia Annua leaves extract and DOX for the treatment of the DOX-induced hepatic toxicity in rats. The authors did a quantitative phytochemical study as well as GS-MS techniques for analyzing phytochemical components of Artemisia Annua leaf extract. Biological evaluations were done using four different groups negative-control, AALE-control, DOX-injected and combination of DOX/AALE group. Overall, I believe that the authors have demonstrated the advantages of Artemisia Annua leaves extract on DOX-induced hepatotoxicity. This manuscript is well written, the science is sound, and the approach solid. In general, this work is useful and should be published in Int. J. Mol. Sci in the present form. 

Author Response

We thank the reviewer for your paragraph about our paper and for his recommendation to our work to be published in Int. J. Mol. Sci in the present form. 
